# The clinical significance of inflammatory biomarkers, IL6 cytokine, and systemic immune inflammatory index in rabbit model of acute and chronic Methicillin-resistant *Staphylococcus epidermidis*-induced osteomyelitis

**Diana-Larisa Ancuța**[1¤]*, **Arianna Barbara Lovati**[2☯], **Cristin Coman**[1☯]

**1** Cantacuzino National Medical Military Institute for Research and Development, Bucharest, Romania,
**2** IRCCS Ospedale Galeazzi - Sant'Ambrogio, Milan, Italy

☯ These authors contributed equally to this work.
¤ Current address: Cantacuzino National Medical Military Institute for Research and Development, Preclinical Testing Unit, Bucharest, Romania
* diana.larisa.ancuta@gmail.com

## Abstract

Infections are a major complication of open fractures and fracture fixation. In this study, an innovative bioactive medical device was used to experimentally treat MRSE-induced osteomyelitis in rabbit tibia. This paper investigates the clinical significance of inflammatory biomarkers (NLR, PLR, MLR and PMR), SII and IL-6 and assesses their role in the development of osteomyelitis. The main objective is to identify the utility of hematological reports derived from neutrophils, leukocytes, monocytes and platelets in the evolution of implant-related osteomyelitis and the estimation of treatment efficiency. In particular, this study compares the response of these inflammatory markers to different treatments in the presence or absence of bioactive materials and/or topical antibiotics over time. The analysis of the threads showed that NLR, PLR and SII had high values in the acute phase of the disease, so that after chronization, they decrease. The animals treated with vancomycin nano-functionalized peptide-enriched silk fibroin-coated implants showed lower levels of inflammatory biomarkers compared to the other groups (empty implants and peptide-enriched silk fibroin-coated implants). NLR, PLR and SII, complemented by IL-6 can be used as fairly accurate biomarkers for the diagnosis of osteomyelitis.

## Introduction

Osteomyelitis is defined as a bone disease caused by microorganisms characterized by inflammation and bone loss. The classification of osteomyelitis, on a broad scale, includes the acute and the chronic form, the delimitation between the two is the time interval that has elapsed

**Data Availability Statement:** The minimal data set of our study can be accessed at the link: https://osf.io/bvawf/?view_only=f71c99de4f204708a478ee79d22bb5a9.

**Funding:** This work was supported by a grant of the Romanian National Authority for Scientific Research and Innovation, CCCDI-UEFISCDI, project number 89/2019 within PNCDI III. This work was also financed under the frame of EuroNanoMed III, ANNAFIB project (JTC2018 058), by the Italian Ministry of Health "Ricerca Corrente" and by the Executive Agency for Higher Education, Research, Development and Innovation Funding (UEFISCDI). The funders had no role in study design, data collection and analysis, decision to publish, or preparation of the manuscript.

**Competing interests:** The authors declare no conflict of interest.

since the contact of the bone with the pathogens. Thus, acute osteomyelitis appears after a few days or weeks, and early diagnosis and treatment have mostly positive outcomes [1]. The chronic form evolves over a longer period, it is accompanied by bone loss and sequestra due to poor vascularization and necrosis, which makes it difficult to transport antimicrobial agents to the focus of osteomyelitis [2]. Acute and chronic onsets are related not only to the duration of onset but also to the intensity of the cellular response.

The common treatment of these conditions involves debridement combined with systemic antibiotics, but taking into account the cumulative capacity of antibiotics at the level of different organs and their possible toxic potential, an ideal solution to solve osteomyelitis is the creation of biomaterials that release local medication. Consequently, several researchers studied the phenomenon and developed a multitude of local antibiotic deliverers in order to prevent the development and proliferation of bacteria at the implant level [3]. Osteomyelitis depends on the nature of the pathogens and the factors associated with the host. The capacity of bacteria of spreading or forming biofilm or evading the immune system together with an immunodeficient background, comorbidities, or emergency situations (trauma) constitute favorable elements for the development of osteomyelitis. The incidence of osteomyelitis associated with implantable devices varies from 30% in the case of open fractures [4] to 3-8% in the case of arthroplasty [5], despite all measures being used to prevent contamination.

Specific signs of inflammation and infection such as fever, redness, pain, and purulent accumulations allow for an easy diagnosis of acute osteomyelitis. Otherwise, chronic or subclinical forms progress with subtle and non-specific symptoms. This requires the establishment of a heterogeneous protocol, including hematological, biochemical and microbial tests, evaluation of the C-reactive protein, and imaging or histopathological examinations. Currently, the gold standard for the diagnosis of osteomyelitis is the bone biopsy followed by microbiological and histopathological examinations of the sample [6, 7].

With respect to the diagnosis of periprosthetic joint infection, there is no test with absolute accuracy. Therefore, the Society of Musculoskeletal Infections (MSIS) and the Society of Infectious Diseases (IDSA) issued 2018 a globally accepted definition that improved confidence in the diagnosis, and led to an efficient therapeutic approach. This definition includes corroborated analysis of clinical data, peripheral blood laboratory results, microbiological culture, histological evaluation of periprosthetic tissues, and intraoperative findings [8].

Blood tests provide important information about inflammatory markers such as erythrocyte sedimentation rate (ESR) and C-reactive protein (CRP). Sometimes, in the presence of subclinical osteomyelitis, normal ESR and CRP values can be found. A proportion of patients presenting with osteomyelitis have normal inflammatory markers at presentation. The ESR threshold of 60 mm/h demonstrated a sensitivity of 74% (95% confidence interval [CI], 67–80) and a specificity of 56% (95% CI, 48–63) for osteomyelitis, while the CRP threshold of 7.9 mg/dL had a sensitivity of 49% (95% CI, 41–57) and a specificity of 80% (95% CI, 74–86). While ESR is better at ruling out osteomyelitis initially, CRP helps distinguish osteomyelitis from soft tissue infection in patients with elevated ESR.

Further prospective studies addressing the prognostic value of ESR and CRP are needed, and a more comprehensive diagnostic algorithm should be developed that includes other diagnostic tests [9]. However, according to one last meta-analysis, the diagnostic value of these markers is limited [10]. In addition, markers related to coagulation (plasma fibrinogen and D-dimer) were found to be effective in diagnosing inflammation. Platelets and average platelet volume also play an essential role in the inflammatory process [11, 12], as well as the neutrophil/lymphocyte ratio (NLR) in the peripheral blood or the monocyte/lymphocyte ratio (MLR) [13].

The complete hematological examination, routine or performed before surgical intervention, can provide additional information on the patient's state of health, if minimal 100 attention is paid to the ratio of blood elements. It can enhance diagnostic accuracy without additional costs. Data from the literature show that NLR, MLR, PLR (platelet/leukocyte ratio), or PMR (platelet/average platelet volume ratio) are associated with the inflammatory and/or infectious state in the body. In fact, systemic changes in NLR, PLR, and MLR represents primary responses to early inflammation and infection. Thereafter, chronic inflammation is caused by the persistence of the inflammatory inducer, such as microbial/viral invasion or a physical injury. Thus, the report by Djordjevic et al [14] claims that the highest levels of MLR and PLR were found in patients with negative blood culture and the lowest in patients with Gram-positive blood culture, an aspect also supported by Naess et al [15] who claim that higher values of NLR and MLR indicated higher probabilities of bacterial infection and low probabilities of viral infection.

Also, Forget et al [16], through their research, demonstrated that very high levels of NLR are frequently associated with acute and persistent inflammatory states. More importantly, the Systemic Immune Inflammatory Index (SII) has been recently used as a prognostic marker in several clinical fields (cancer, infections, surgeries, etc.) and can be used to diagnose infections in the absence of clear signs. The SII is calculated as (NEU × PLT)/LYM (NEU, PLT, and LYM represent neutrophil counts, platelet counts, and lymphocyte counts, respectively). Compared to NLR, MLR, and PLR biomarkers, the SII describes the host's immune system imbalance and inflammatory condition. Even if various studies have demonstrated the correlation between NLR, PLR, and many diseases such as inflammatory diseases [17] as well as a correlation between high NLR and PLR, which can guide the diagnosis of a bone infection such as osteomyelitis [18], the diagnostic value of NLR, MLR and PLR in anticipating bone infections and their prognosis remains to be explored. This suggests that there is an urgent need for new inflammatory markers to diagnose osteomyelitis. Among inflammatory cytokines, interleukin (IL) -6 has recently been identified as a potential target for inflammatory disease. Some studies revealed that IL-6 is an active player in the immune response and showed that excess of this cytokine and/or its receptor contributes to the pathogenesis of the inflammatory process [19]. Other studies demonstrate that bacterial challenge of osteoblasts during bone diseases, such as osteomyelitis, induces cells to produce inflammatory molecules that can direct appropriate host responses or contribute to progressive inflammatory damage and bone destruction [20–22].

In this scenario, it could be useful to analyze these inflammatory markers in the course of infections, where there is an alteration of the oxidative stress mediators, although an inflammatory status is not clinically evident yet. Since there are no data related to these analyzes in animals, we proposed to evaluate these reports and to determine if there is a modification of the inflammatory markers, IL-6 and SII in a rabbit model of osteomyelitis treated or not with locally delivered antibiotics, as the most frequently used, relevant and reproducible model of orthopedic infections [23]. Thus, special attention is given to experimental laboratory animals, which researchers used to study human osteomyelitis. Indeed, *in vivo* models are the ideal solution for the study of advanced diagnostic and therapeutic approaches to translate into clinical practice.

## Materials and methods

### Animals, bacterial strains, and medical devices used in the experiments

For each model (acute and chronic), 54 adult New Zealand White rabbits, male and female, with an average weight of 3000 grams were used. The animals were ordered from the CI

animal facility and were housed in individual cages with 12 h light/12h dark cycles. They were acclimatized for five days under the same experimental conditions, at a temperature of 16-21˚C and relative humidity of 45-65%, during which time they received food and water ad libitum.

Methicillin-resistant *Staphylococcus epidermidis* (MRSE, GOI1153754-03-14)—isolated from a knee prosthesis of a human patient—was received from the Istituto Ortopedico Galeazzi of Milan (Italy) [24] and processed in the CI Microbiology Laboratory—the 164 concentration of MRSE used in the study was $5x10^{10}CFU/ml$. Humans are not directly involved in the study. Indeed MRSE, GOI1153754-03-14 strain has been isolated during common diagnostic procedures and the ethical committee approval is not required for diagnostic activities. No human tissue samples are employed but a bacterial strain detached from prosthetic metal implant, and that the bacterial strain is no more attributable to a specific patient.

Medical devices tested: empty titanium metal implants (TMMB10, Zimmer Biomet), titanium implants coated with peptide-enriched silk fibroin, and titanium implants coated with vancomycin nano-functionalized peptide-enriched silk fibroin, as developed elsewhere [25].

## Experimental design and procedures

For each study, 54 rabbits, males and females in equal number, were weighed and marked on the ear with an animal marker, assigning an individual ID. Furthermore, depending on weight, sex, clinical status (healthy—acute study, with osteomyelitis—chronic study), 3 uniform groups (9 males and 9 females) were created, depending on the tested implant. The animals were deeply anesthetized by neuroleptanalgesia using a mixture of ketamine (50mg/kg IM, Vetased, Farmavet, Romania) and acepromazine (1mg/kg IM, Sedam, Farmavet, Romania). In the proximal area of the tibia, at approximately 8 mm from the tibio-femoral-patellar joint at the level of the left hind limb, a bone defect was created using progressive diameter drills (ø 1-3.5 mm), under cooling continue with sterile saline solution.

To induce acute osteomyelitis, implants (M, C and T) previously immersed in the bacterial suspension were inserted in the bone defects. Before closing the wound, 0.1 ml of MRSE at the same concentration was dispersed over the implants. The periosteum, muscles, and skin were sutured with absorbable thread 2/0 (Megasorb, Buritis). During the follow-up period (14 days), the rabbits were monitored clinically and hematologically, according to the protocols described elsewhere [26].

To induce the chronic osteomyelitis, in the first stage, a gauze mesh immersed in the bacterial suspension was inserted into the defects created at the level of the tibia, over which an additional 0.1 ml of MRSE at the same concentration was inoculated and the wound was closed. 14 days after the bacterial inocula, further surgery was carried out to remove gauze tampons and to place implants (M, C, and T) at the infected site. This approach was to create a chronic osteomyelitis support, an environment in which the implants could be tested [27].

Post-operatively, the rabbits received analgesic treatment (Ketoprofen—3mg/kg SC, Dopharma, Romania) for 3 days. During the follow-up period (60 days), the rabbits were monitored clinically and hematologically. At the end of the study, the animals were euthanized by an anesthetic overdose, and the tibias were taken for histological analysis.

## Blood analyses: Inflammatory markers and systemic inflammatory index

Blood samples were collected in EDTA-preconditioned tubes (Sarstedt, Germany) from the auricular vein on days 0 (basal healthy values), 1, 3, 7, and 14 for acute osteomyelitis and on days 0, 21, 43, and 60 for chronic osteomyelitis. In order to achieve a complete and correct hematological examination, the samples were analyzed at an interval of about one hour after

collection. The hematological examination was carried out with the ProCyte Dx analyzer (IDEXX Laboratories). The reference intervals are represented by the basal values at day 0. Therefore, based on the results obtained at the hematological examination, we calculated the ratios between neutrophil and lymphocyte (NLR = NEU/LYM), monocyte and lymphocyte (MLR = MONO/LYM), platelet and lymphocyte (PLR = PLT/LYM), platelets and the average platelet volume (PMR = PLT/MPV). Moreover, the systemic immune-inflammatory index and the cytokine IL-6 were also analyzed. The interleukin 6 (IL-6) assay was conducted using plasma samples obtained from the blood collected in Lithium-Heparin preconditioned tubes and the measurement of interleukin 6 (IL-6) was performed using an enzyme-linked immuno-sorbent assay (ELISA) method. The specific type and brand of the kit used for measuring IL-6 was R&D Systems™ Rabbit IL-6 DuoSet ELISA (Fisher Scientific, Germany) and the manufacturer's instructions were followed for the processing of the samples.

## Statistical analysis

The sample size was calculated a priori through a two-sample t-test with $\alpha$ error = 0.05% and 80% power (G*Power 3.1 software, Dusseldorf, Germany) [28]. The statistical analysis was performed with GraphPad Prism 5 software (GraphPad Software, San Diego, California, USA). The variables taken into account were related to the group of animals (acute and chronic), depending on the medical device tested treated or not (M, C, and T), the days of clinical monitoring, as well as the laboratory data (white blood cells, neutrophils, lymphocytes, platelets, monocytes, and IL-6).

The inclusion criteria, in the case of the acute study, referred to the uniform formation of groups of animals, in terms of weight and sex. In the case of the chronic study, in addition to the inclusion criteria from the acute study, we also added the criterion of clinical expression of osteomyelitis. In both studies, the exclusion criteria followed the elimination of animals that did not express specific signs of osteomyelitis or that presented fractures of the operated limbs, weight loss greater than 25%, lack of food consumption for more than 48 hours. After checking the normal distribution of the data using the Shapiro-Wilk test, the intergroup comparisons were analyzed with a one-way analysis of variance (ANOVA) and then coupled with Bonferroni's post hoc test. All data are reported as means ± standard error (SE). The level of significancewas $p < 0.05$.

For each study, 6 different investigators were involved as follows: a first investigator anesthetized the animals based on the randomization weight table. A second investigator was responsible for the surgical procedure, postoperative treatment and clinical monitoring. This investigator was the only person who knew the treatment group allocation. 2 investigators were responsible for the blind processing of the biological samples collected from the animals and 2 more investigators, who analyzed the raw data and issued the final conclusions.

## Results

Various inflammatory factors have been described as prognostic biomarkers, such as the neutrophil–lymphocyte ratio (NLR), systemic immune-inflammation index (SII) and IL-6 cytokine.

In our experimental setting, we found a time-related variation of all the evaluated parameters. In the acute phase (D14), the NLR ratio (Fig 1) is significantly different from that of the baseline (basal healthy values at D0—red dotted line) in all the groups (M $p < 0.001$, C $p < 0.001$, and T $p < 0.01$, respectively).

This trend is also confirmed by the analysis of the SII values (Fig 1), as a more comprehensive parameter of whole-body inflammation. Similarly, in M and C groups at D60, both NLR

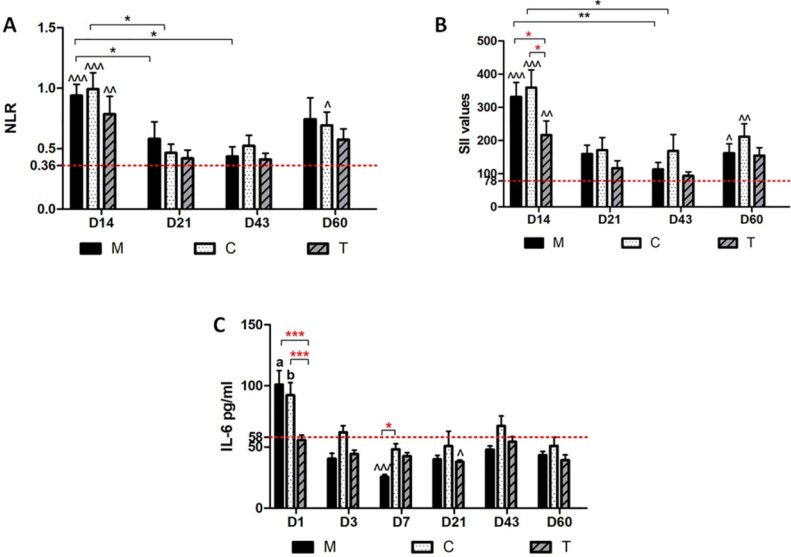

**Fig 1. Histograms for NLR, SII and IL-6 cytokine.** Histograms reporting (A) the neutrophil–lymphocyte ratio (NLR), (B) the systemic immune-inflammation index (SII) and (C) IL-6 cytokine of the three differently treated groups (M, C, and T) over time. Statistical difference of the group vs healthy values (red dotted lines); (* red) () statistical differences among groups and (*black) time points; (a) $p < 0.001$ between group M at D1 and D3, 7, 21 and 60; (b) $p < 0.01$ between group C at D1 and D3, 7, 21 and 60.

and SII increased significantly from to the basal values. Moreover, the SII at D14 showed a difference among groups. Indeed, the treated group (T) had significantly lower SII values than M (empty implant) and C (implant coated with silk fibroin w/o antibiotic) for $p < 0.05$.

In the chronic phase, the host's immune response and inflammatory markers are expected to decline. However, at the latter time point (D60), in the presence of a consistent bacterial infection, both the immune response and the inflammatory biomarkers increased again, as demonstrated in our histogram trend, in which this trend was confirmed for the M and C groups but not for the T group. Therefore, this aspect underlines the MRSE characteristic of maintaining the infection at a subclinical level and its reactivation at long intervals after the acute clinical manifestation, in the absence of a long-acting treatment. In the case of our study, this idea proves the effectiveness of the treatment in controlling the inflammatory response through its antimicrobial activity against bacterial proliferation.

Cytokine IL-6 plays a protective role in immune responses to bacterial infections. Our data showed that IL-6 was involved at the very beginning (D1) of the bacterial infection (Fig 1). Indeed, IL-6 is the major inductor of the acute phase proteins and is involved in the control of neutrophil and monocyte responses following infection, thus confirming the increase of NLR and SII at D14. Once again, the IL-6 values showed a difference among groups, where the treated group (T) had significantly lower IL-6 values compared to M (empty implant) and C (implant coated with silk fibroin w/o antibiotic) for $p < 0.001$, thus supporting the SII trend at D14.

During the chronic phase, IL-6 plays a marginal role, in fact, the IL-6 values decreased significantly over time in groups M and C. More importantly, group T did not show any increased difference compared to the basal values at any time point. However, in the case of a chronic bacterial infection, IL-6 can fluctuate and lead to enhanced systemic inflammation associated with the host response. In this series, this phenomenon was found at D43 with an

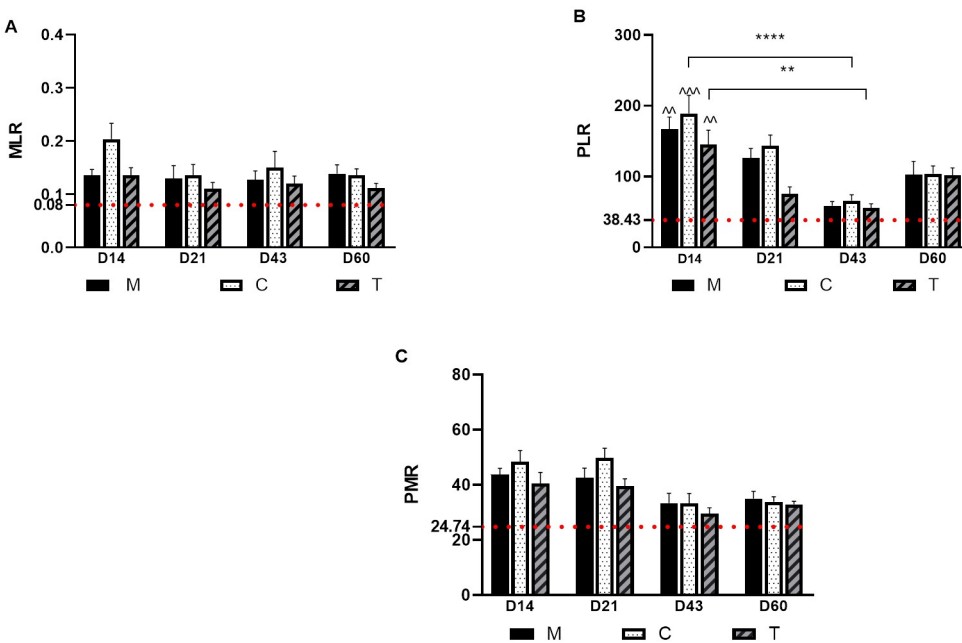

**Fig 2. Histograms for MLR, PLR and PMR.** Histograms reporting (A) the monocyte-lymphocyte ratio (MLR), (B) the platelet-lymphocyte ratio (PLR), and (C) the platelet-mean platelet volume ratio (PMR) of the three differently treated groups (M, C, and T) over time. (*) statistical difference of the groups (*for $p < 0.05$, ** for $p < 0.001$, and **** for $p < 0.0001$). The red dotted lines are reporting the mean of basal values of the ratios MLR, PLR, and PMR, statistical difference of thehealthy values (red dotted lines) vs groupˆ.

increase in IL-6 in all the groups followed by the increase of NLR and SII at D60 in groups M and C.

The MLR histogram (Fig 2) did not show any significant differences over time and groups, only a statistical difference was observed between groups C and T on D14 ($p = 0.04$) and on day 60 ($p = 0.02$). With respect to PLR (Fig 2), there were significant differences at day 14, in the case of all groups compared to the basal values (dotted red line). As in the case of NLR and SII, the PLR ratio tends to decrease as the disease becomes chronic, and with a slight increase observed at D60. The PMR values (Fig 2) remained high on D14 up to D21, and then decreased near to the basal values in the chronic phase of the disease. On D60, the ratio also rose slightly. Significant differences were found in group C on D21 compared to D43 ($p < 0.001$) and compared to groups M and T within the same time interval ($p = 0.05$).

## Discussion

The study is the first to investigate the clinical significance of inflammatory biomarkers (NLR, PLR, MLR and PMR), the SII, and IL-6 in a well-established rabbit model of orthopedic infections to evaluate the progression of chronic osteomyelitis development.

The main objective of this study was to assess the utility of hematological data derived from neutrophils, leukocytes, monocytes, and platelet counts in tracking the progression of implant-associated osteomyelitis and evaluating treatment in an experimentally MRSE-induced bone infection. In particular, this study compares the response of these inflammatory markers to various treatments with or without locally deliered bioactive materials and/or antibiotics, over time. To our knowledge, these hematological indices have not been previously studied together for this purpose. The main pathogen responsible for osteomyelitis in humans

is *Staphylococcus aureus* (SA), but other bacteria such as *Staphylococcus epidermidis*, *Staphylococcus lugdunensis*, and *Propionibacterium acnes* [29] can also cause the disease [30]. The biofilm produced by these bacteria can complicate antibiotic treatment. Selecting an appropriate *in vivo* model of osteomyelitis is crucial due to the varying advantages and limitations of different animal models. While rodents are the extensively studied, rabbits are commonly used in musculoskeletal research and were the first documented animal model for osteomyelitis [3]. Rabbits are reliable and reproducible models for orthopedic infections due to their bone structure, immune system, and susceptibility to infections, which closely resemble those of humans [31, 32]. Furthermore, their size allows for the testing of medical devices, such as implants, in the presence of bacterial contaminations [23].

The induction of osteomyelitis in rabbits can be achieved by creating mechanical fractures or bone defects, which may be left untreated or filled with materials that promote bacterial adhesion, [33] or by using contaminated implants [34]. The development of experimental osteomyelitis can be evaluated through clinical examination, radiography, bone biopsy, and bacterial cultures. Studies by Odekerken et al. [35] showed that F-FDG micro-PET is a sensitive tool for early detection of osteomyelitis, and PCR can identify bacteria even at low or metabolically inactive levels [35, 36].

Research on the lagomorph immune system focuses on *Oryctolagus cuniculus*, the reference model for immunology studies. Investigations into diseases such as myxomatosis or rabbit hemorrhagic disease have significantly expanded our understanding of the human oncological field. Advances in understanding innate immunity include the identification of interleukins, chemokines and their receptors, Toll-like receptors, antiviral proteins (RIG-I and Trim5), and genes encoding fucosyltransferases, which rabbit hemorrhagic disease virus uses to invade respiratory and intestinal epithelial cells. Regarding adaptive immunity, the major histocompatibility complex (MHC) and T cells are notable for their genetic diversity in the loci encoding the variable and constant regions of antibodies [37]. Our research suggests that longitudinal analysis of human osteomyelitis can be effectively conducted using cost-efficient methods by focusing on the ratios of white blood cells and inflammatory cytokines. Given the lack of literature on the repetitive analysis of inflammatory markers (NLR, MLR, PLR, PMR), we found it appropriate to track these markers in a study of experimental osteomyelitis in rabbits with MRSE during both the acute and chronic of the disease. Our findings indicate that NLR, PLR, and SII significantly increased during the acute onset of osteomyelitis, as expected. In the chronic phase, the immune response and inflammatory markers typically decline. However, at a later time point (D60), persistent bacterial infection caused both the immune response and inflammatory biomarkers to rise again, resulting in subclinical low-grade inflammation. Specifically, in our series, we demonstrated that animals treated with vancomycin nano-functionalized peptide-enriched silk fibroin-coated implants (group T) exhibited lower NLR, PLR and SII values compared to the other groups during both the acute or chronic phases. Otherwise, MLR and PMR did not show significant values for diagnosing osteomyelitis or evaluating the effectiveness of the treatment.

The systemic inflammatory response triggered by bacterial contamination determines changes in neutrophil, lymphocyte, monocyte and platelet counts. During infections, the immune system induces an inflammatory response to fight pathogens, balancing pathogen clearance with minimizing host tissue damage, ultimately leading to tissue recovery [38]. In fact, neutrophils are the first responders to infection, while lymphocytes generate the adaptive immune response [14, 39, 40]. Consequently, NLR and SII can provide important information about the onset, persistence of infection, and even the occurrence of sepsis. In a cohort study by Salciccioli et al. [41], NLR was significantly higher in patients with sepsis compared to patients without sepsis. Similarly, elevated values are observed in bacteriemia studies, as

described by de Jager et al., where patients with blood cultures contaminated with Gram-negative bacteria had higher NLR than those with Gram-positive pathogens [42]. In contrast, data indicate that lymphocytes and monocytes exhibit higher values in patients with Gram-positive blood cultures compared to those with polymicrobial or negative blood cultures [14]. Emektar et al. observed a elevated PLR levels in patients who had experienced trauma, such as hip fracture or acute appendicitis in children, suggesting that this biomarker is closely related to persistent inflammation [43]. Although recent investigations have proposed PMR ratio as a strong diagnostic marker of sepsis [43], our data do not support this findings. Differently, it has been demonstrated that platelet activation plays a role in the pathogenesis of thrombosis associated with *Staphylococcus aureus* osteomyelitis [44], confirming the importance of platelet increase during both acute and chronic osteomyelitis. Additionally, while inflammatory cytokines like IL-6 are protective in immune responses to bacterial infections, they also promote osteoclast differentiation and inhibit osteoblast formation [45]. Moreover, some studies have demonstrated local production of IL-6 at bacterial infection sites in human osteomyelitis associated with S. aureus [21], as well as in a murine model [46]. In addition, bacterial-stimulated osteoblasts express IL-6 and IL-12, which induce an immune response by attracting leukocytes [22]. Our data indicate that the release of IL-6 occurs primarily at the very acute stage of the disease, but this cytokine levels drop rapidly within 48-72 hours. In summary, the inflammatory biomarkers analyzed currently do not provide conclusive data for validation. However, our study observed increased levels of all ratios, especially in acute osteomyelitis. A possible explanation is that these ratios lose their prognostic capacity once the disease becomes chronic, as reported by Oh et al. [47], where all analyzed biomarkers showed low levels after 48-72 hours [48]. This suggests that the potential value of each inflammatory marker should not be assessed independently but rather evaluated collectively to detect inflammatory markers associated with osteomyelitis.

There were some several limitations in our study. Firstly, the literature lacks reported threshold values for inflammatory markers in animal models, particularly in rabbits, making comparisons difficult. Moreover, unlike human blood counts often performed on hospitalized patients, our measurements were taken at specific, well-defined intervals throughout the experiment. Our findings accentuate the necessity for further research to establish standardized cutoff values for these inflammatory markers in both human and animal models to effectively diagnose and prognose chronic and/or subclinical bacterial infections. Another limitation was the absence of a control group undergoing surgery without bacterial infection, as it is well known that surgery alone can alter inflammatory markers (leukocytes, platelets and cytokines) particularly during the early postoperative period. This may have influenced the data obtained during the acute phase of osteomyelitis. Additionally, our study did not investigate CRP and ESR, which could corroborate the degree of inflammation detected through the analyzed parameters. However, Ustundag and colleagues demonstrated that SII and PLR values might be higher in low-grade inflammation patients with mildly elevated CRP, indicating that CRP did not correlate with SII and PLR [49].

## Conclusion

In our study, we found that NLR, PLR, and SII increased significantly during the acute onset of osteomyelitis, as expected. This aspect was also supported by the rapid increase of IL-6 in the first hours of the infection. In the chronic phase, when it was expected that the inflammatory markers would decrease, we observed a slight increase in them (D60) because of the high bacterial load.

The animals treated with vancomycin nano-functionalized peptide-enriched silk fibroin-coated implants had lower NLR, PLR, and SII values compared to the other groups either in the acute or chronic phase. NLR, PLR and SII associated with the clinical context may signal the presence of an osteomyelitis-type infection, especially in the acute phase. However, more long-term studies on the same samples are needed to demonstrate the ability of these inflammatory biomarkers to anticipate the chronic phases of osteomyelitis.

## Acknowledgments

We would like to thank biochemist Petronica Gheorghiu for the hematological analyzes performed and also thank Dr. Iuliana Caras for the immunological analysis.

## Author Contributions

**Conceptualization:** Cristin Coman.

**Formal analysis:** Diana-Larisa Ancuţa, Arianna Barbara Lovati, Cristin Coman.

**Funding acquisition:** Arianna Barbara Lovati.

**Investigation:** Diana-Larisa Ancuţa, Arianna Barbara Lovati, Cristin Coman.

**Methodology:** Cristin Coman.

**Project administration:** Arianna Barbara Lovati, Cristin Coman.

**Software:** Diana-Larisa Ancuţa.

**Supervision:** Arianna Barbara Lovati, Cristin Coman.

**Validation:** Arianna Barbara Lovati, Cristin Coman.

**Visualization:** Arianna Barbara Lovati, Cristin Coman.

**Writing – original draft:** Diana-Larisa Ancuţa.

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
