## [Decision Letter · Decision Letter 0]

27 May 2024

PONE-D-24-02543The clinical significance of inflammatory biomarkers, IL-6 cytokine, and systemic immune inflammatory index in rabbit model of acute and chronic Methicillin-resistant Staphylococcus epidermidis-induced osteomyelitisPLOS ONE

Dear Dr. Larisa,

Thank you for submitting your manuscript to PLOS ONE. After careful consideration, we feel that it has merit but does not fully meet PLOS ONE’s publication criteria as it currently stands. Therefore, we invite you to submit a revised version of the manuscript that addresses the points raised during the review process.

We look forward to receiving your revised manuscript.

Kind regards,

Seyed Mostafa Hosseini

Academic Editor

PLOS ONE

Journal Requirements:

 [This work was supported by a grant of the Romanian National Authority for Scientific Research and Innovation, CCCDI-UEFISCDI, project number 89/2019 within PNCDI III. This work was financed under the frame of EuroNanoMed III, ANNAFIB project (JTC2018 058), by the Italian Ministry of Health and by the Executive Agency for Higher Education, Research, Development and Innovation Funding (UEFISCDI).].  

4. We note that your Data Availability Statement is currently as follows: [All relevant data are within the manuscript.]

Reviewers' comments:

Reviewer's Responses to Questions

**Comments to the Author**

1. Is the manuscript technically sound, and do the data support the conclusions?

Reviewer #1: Yes

Reviewer #2: Yes

2. Has the statistical analysis been performed appropriately and rigorously? 

Reviewer #1: I Don't Know

Reviewer #2: Yes

3. Have the authors made all data underlying the findings in their manuscript fully available?

Reviewer #1: Yes

Reviewer #2: Yes

4. Is the manuscript presented in an intelligible fashion and written in standard English?

Reviewer #1: Yes

Reviewer #2: Yes

5. Review Comments to the Author

Reviewer #1: 1- In line 203, the selection of sampling days for hematological tests is based on what and how?

2- In line 213, What sample was used in interleukin 6 assay?

3- In line 213, What was the measurement method, type and brand of the kit used to measure interleukin 6?

Reviewer #2: The study presents a good balance of scientific rigor and methodology. The use of a rabbit model of acute and chronic osteomyelitis induced by MRSE is a relevant and well-established model. The inclusion of different treatment groups, including empty titanium metal implants, peptide-enriched silk fibroin-coated implants, and vancomycin nano-functionalized peptide-enriched silk fibroin-coated implants, provides a good basis for comparing the effects of different treatments.

The study measures a range of inflammatory biomarkers, including NLR, PLR, MLR, PMR, SII, and IL-6, which are relevant to the development of osteomyelitis. The analysis of these biomarkers is performed using routine blood tests, which is a practical and clinically relevant approach.

However, there are some limitations to the study. The sample size is relatively small, with only 54 rabbits used in each study. This may not provide sufficient statistical power to detect significant differences between the treatment groups. Additionally, the study only measures the inflammatory biomarkers at a single time point, without following the rabbits over time to see how these biomarkers change in response to treatment.

Writing

The writing is generally clear and concise, with good use of headings and subheadings to organize the text. The introduction provides a good background on the clinical significance of osteomyelitis and the importance of developing new diagnostic markers.

However, there are some areas where the writing could be improved. The transitions between sections could be smoother, and some sentences could be rephrased for clarity. Additionally, the text could benefit from more precise language and fewer technical jargon.

Structure

The structure of the article is generally good, with a clear introduction that sets out the background and aims of the study. The methods section is well-organized and provides a clear description of the experimental design and procedures.

The results section is well-organized and presents the data in a clear and concise manner. However, some of the tables could be improved by adding more detail or context to help readers understand the data.

The discussion section is generally well-organized and provides a good summary of the main findings. However, some of the sections could be condensed or reorganized to improve the flow of the text.

Usefulness of hematological reports: The study evaluates the usefulness of hematological reports derived from neutrophils, leukocytes, monocytes, and platelet formula regarding the evolution of implant-associated osteomyelitis and treatment evaluation. The discussion section highlights the importance of these inflammatory biomarkers in diagnosing and monitoring osteomyelitis.

Longitudinal analysis of inflammatory markers: The study follows the inflammatory biomarkers (NLR, MLR, PLR, PMR, SII, and IL-6) over time to understand their response to different treatments and to identify potential changes in the chronic phase of osteomyelitis. The discussion section suggests that the longitudinal analysis of these markers is crucial for understanding the evolution of osteomyelitis.

Comparison of different treatments: The study compares the response of inflammatory biomarkers to different treatments, including empty titanium metal implants, peptide-enriched silk fibroin-coated implants, and vancomycin nano-functionalized peptide-enriched silk fibroin-coated implants. The discussion section highlights the importance of comparing different treatments to evaluate their effectiveness in treating osteomyelitis.

Role of platelets in inflammation: The study investigates the role of platelets in inflammation and suggests that platelet augmentation is important during acute and chronic osteomyelitis.

However, some limitations of the study are mentioned:

1. Lack of standardized cutoff values: The study notes that threshold values for inflammatory markers have not been reported in animal models, especially in rabbits, and that further research is needed to establish standardized cutoff values.

2. Limitations of the study design: The study mentions that the absence of a test group undergoing surgery without bacterial infection may have impacted the data obtained during the acute phase of osteomyelitis.

3. Limited investigation of CRP and ESR: The study notes that CRP and ESR were not investigated, although they could corroborate the degree of inflammation detected through the analyzed parameters.

Overall, the article presents a good balance of scientific rigor and methodology, with some areas where the writing could be improved. With some revisions to address these areas, the article could be strengthened further.

6. PLOS authors have the option to publish the peer review history of their article (what does this mean?). If published, this will include your full peer review and any attached files.

Reviewer #1: No

Reviewer #2: **Yes: **Dr SeyedMousa Motavallihaghi

---

## [Author Response · Author response to Decision Letter 0]

7 Jun 2024

Dear editors and reviewers,

Thank you for your suggestions, comments and support in publishing our article. We have now revised the manuscript and answered exactly what you requested, as follows:

REV 1

1- In line 203, the selection of sampling days for hematological tests is based on what and how?

The authors thank the reviewer for the interesting question. The selection of sampling days for hematological tests was based on different stages of the induced osteomyelitis, both acute and chronic. For acute osteomyelitis, blood samples were collected on days 0 (baseline), 1, 3, 7, and 14. For chronic osteomyelitis, samples were taken on days 0, 21, 43, and 60. These time points were chosen to capture the progression and resolution of the inflammatory response over time.

2. In line 213, what sample was used in the interleukin 6 assay?

The authors thank the reviewer to notice the lack of information regarding the sample and measurement method used to test IL-6. The interleukin 6 (IL-6) assay was conducted using plasma samples obtained from the blood collected in Lithium-Heparin preconditioned tubes.

3. In line 213, What was the measurement method, type and brand of the kit used to measure interleukin 6?

The measurement of interleukin 6 (IL-6) was performed using an enzyme-linked immunosorbent assay (ELISA) method. The specific type and brand of the kit used for measuring IL-6 was R&D Systems™ Rabbit IL-6 DuoSet ELISA (Fisher Scientific, Germany) and the manufacturer's instructions were followed for the processing of the samples. This information and the previous one have been added to the revised version of the manuscript at lines 178-184.

REV 2

The study presents a good balance of scientific rigor and methodology. The use of a rabbit model of acute and chronic osteomyelitis induced by MRSE is a relevant and well-established model. The inclusion of different treatment groups, including empty titanium metal implants, peptide-enriched silk fibroin-coated implants, and vancomycin nano-functionalized peptide-enriched silk fibroin-coated implants, provides a good basis for comparing the effects of different treatments.

The study measures a range of inflammatory biomarkers, including NLR, PLR, MLR, PMR, SII, and IL-6, which are relevant to the development of osteomyelitis. The analysis of these biomarkers is performed using routine blood tests, which is a practical and clinically relevant approach.

Thank you for your thorough and insightful review. We appreciate your positive feedback on the scientific rigor, methodology, and relevance of our study, as well as your constructive comments on areas for improvement. Below are our responses to your specific points:

However, there are some limitations to the study. The sample size is relatively small, with only 54 rabbits used in each study. This may not provide sufficient statistical power to detect significant differences between the treatment groups. Additionally, the study only measures the inflammatory biomarkers at a single time point, without following the rabbits over time to see how these biomarkers change in response to treatment.

Sample Size and Statistical Power:

We acknowledge that the sample size of 54 rabbits per study is relatively small. The sample size was calculated a priori through a two-sample t-test with α error = 0.05% and 80% power, ensuring that it is statistically sufficient to detect significant differences in this context. Future studies with larger sample sizes could provide more robust data and further validate our findings.

Single Time Point Measurement:

We appreciate the suggestion to follow the inflammatory biomarkers over time. In our study, we did measure the inflammatory biomarkers at multiple time points (days 0, 1, 3, 7, and 14 for acute osteomyelitis and days 0, 21, 43, and 60 for chronic osteomyelitis), as detailed in the manuscript. This longitudinal approach was intended to capture the dynamic changes in biomarkers in response to treatment over time. We will ensure that this aspect is more clearly highlighted in the revised manuscript.

Writing

The writing is generally clear and concise, with good use of headings and subheadings to organize the text. The introduction provides a good background on the clinical significance of osteomyelitis and the importance of developing new diagnostic markers.

However, there are some areas where the writing could be improved. The transitions between sections could be smoother, and some sentences could be rephrased for clarity. Additionally, the text could benefit from more precise language and fewer technical jargon.

The authors revised the manuscript to improve the transitions between sections, rephrase sentences (especially in the discussion section) for clarity, and reduce technical jargon where possible. Our goal is to make the text more accessible while maintaining scientific accuracy.

Structure

The structure of the article is generally good, with a clear introduction that sets out the background and aims of the study. The methods section is well-organized and provides a clear description of the experimental design and procedures.

The results section is well-organized and presents the data in a clear and concise manner. However, some of the tables could be improved by adding more detail or context to help readers understand the data.

The discussion section is generally well-organized and provides a good summary of the main findings. However, some of the sections could be condensed or reorganized to improve the flow of the text.

According to the reviewer’s suggestion, the authors reorganized the discussion to improve the flow of the text.

Usefulness of hematological reports: The study evaluates the usefulness of hematological reports derived from neutrophils, leukocytes, monocytes, and platelet formula regarding the evolution of implant-associated osteomyelitis and treatment evaluation. The discussion section highlights the importance of these inflammatory biomarkers in diagnosing and monitoring osteomyelitis.

Longitudinal analysis of inflammatory markers: The study follows the inflammatory biomarkers (NLR, MLR, PLR, PMR, SII, and IL-6) over time to understand their response to different treatments and to identify potential changes in the chronic phase of osteomyelitis. The discussion section suggests that the longitudinal analysis of these markers is crucial for understanding the evolution of osteomyelitis.

Comparison of different treatments: The study compares the response of inflammatory biomarkers to different treatments, including empty titanium metal implants, peptide-enriched silk fibroin-coated implants, and vancomycin nano-functionalized peptide-enriched silk fibroin-coated implants. The discussion section highlights the importance of comparing different treatments to evaluate their effectiveness in treating osteomyelitis.

Role of platelets in inflammation: The study investigates the role of platelets in inflammation and suggests that platelet augmentation is important during acute and chronic osteomyelitis.

However, some limitations of the study are mentioned:

1. Lack of standardized cutoff values: The study notes that threshold values for inflammatory markers have not been reported in animal models, especially in rabbits, and that further research is needed to establish standardized cutoff values.

2. Limitations of the study design: The study mentions that the absence of a test group undergoing surgery without bacterial infection may have impacted the data obtained during the acute phase of osteomyelitis.

3. Limited investigation of CRP and ESR: The study notes that CRP and ESR were not investigated, although they could corroborate the degree of inflammation detected through the analyzed parameters.

Overall, the article presents a good balance of scientific rigor and methodology, with some areas where the writing could be improved. With some revisions to address these areas, the article could be strengthened further.

The authors updated the manuscript to reflect the points mentioned above, ensuring a clearer presentation of our longitudinal analysis of inflammatory markers, improving the clarity and flow of the text, and adding more detailed context to our discussion section. We hope that the revised version of the manuscript could satisfy the reviewer but also readers’ requirements.

---

## [Decision Letter · Decision Letter 1]

7 Aug 2024

The clinical significance of inflammatory biomarkers, IL-6 cytokine, and systemic immune inflammatory index in rabbit model of acute and chronic Methicillin-resistant Staphylococcus epidermidis-induced osteomyelitis

PONE-D-24-02543R1

Dear Dr. Ancuța,

We’re pleased to inform you that your manuscript has been judged scientifically suitable for publication and will be formally accepted for publication once it meets all outstanding technical requirements.

Kind regards,

Seyed Mostafa Hosseini

Academic Editor

PLOS ONE

Additional Editor Comments (optional):

Reviewers' comments:

Reviewer's Responses to Questions

**Comments to the Author**

1. If the authors have adequately addressed your comments raised in a previous round of review and you feel that this manuscript is now acceptable for publication, you may indicate that here to bypass the “Comments to the Author” section, enter your conflict of interest statement in the “Confidential to Editor” section, and submit your "Accept" recommendation.

Reviewer #1: All comments have been addressed

Reviewer #2: All comments have been addressed

2. Is the manuscript technically sound, and do the data support the conclusions?

Reviewer #1: Yes

Reviewer #2: Yes

3. Has the statistical analysis been performed appropriately and rigorously? 

Reviewer #1: Yes

Reviewer #2: Yes

4. Have the authors made all data underlying the findings in their manuscript fully available?

Reviewer #1: (No Response)

Reviewer #2: Yes

5. Is the manuscript presented in an intelligible fashion and written in standard English?

Reviewer #1: Yes

Reviewer #2: Yes

6. Review Comments to the Author

Reviewer #1: (No Response)

Reviewer #2: (No Response)

7. PLOS authors have the option to publish the peer review history of their article (what does this mean?). If published, this will include your full peer review and any attached files.

Reviewer #1: **Yes: **Pezhman Karami

Reviewer #2: No

---

## [Editor Report · Acceptance letter]

19 Aug 2024

PONE-D-24-02543R1 

PLOS ONE

Dear Dr. Ancuța, 

I'm pleased to inform you that your manuscript has been deemed suitable for publication in PLOS ONE. Congratulations! Your manuscript is now being handed over to our production team.

Kind regards, 

on behalf of

Dr. Seyed Mostafa Hosseini 

Academic Editor

PLOS ONE